# Cost of fruiting, size regulation, and senescence in a long-lived geophyte

**James D. Thomson**[1,2*], **David F. Andrews**[3]

**1** Department of Ecology and Evolutionary Biology, University of Toronto, Toronto, Ontario, Canada,
**2** Rocky Mountain Biological Laboratory, Crested Butte, Colorado, United States of America,
**3** Department of Statistical Sciences, University of Toronto, Toronto, Ontario, Canada

\* james.thomson@utoronto.ca

## Abstract

Grown in a transplant garden that provides field conditions but prevents predation by pocket gophers, plants of *Erythronium grandiflorum* (Liliaceae) have been exhumed annually (as dormant corms), photographed, and weighed over 33 years. From seed, plants grow to flowering size in about 5–6 years. They subsequently regulate their size by occasional vegetative splitting and by flowering and fruiting; producing one fruit costs a plant about 8% of the weight it would have gained if it had not flowered. Death is rare: a few plants have gradually lost weight and died in a way consistent with classical senescence, but others have died suddenly from fungal infection after previously growing robustly. Additional years of observation will be needed to clarify the issue of senescence. The data are archived and future collaborators are sought.

## Introduction

Evolutionary ecologists have been understandably drawn to studying monocarpic plants, especially annuals. The monocarpic life history is simple: a period of accumulating resources is followed by a single episode of converting those resources into progeny. Furthermore, a count of those progeny can be defended as a reasonable surrogate of fitness. In contrast, long-lived perennial plants introduce complications arising from the sequential interplay of many periods of growth interspersed with many episodes of sexual reproduction. Emergent phenomena, such as senescence and the cost of sexual reproduction to vegetative growth, have attracted the attention of life-history theoreticians. Early models of resource allocation by Pugliese [1] and Iwasa and Cohen [2] suggested that an optimal developmental program would be for a plant to grow vegetatively to a certain size, then to maintain that vegetative size while putting further resources into sexual reproduction. Such maintenance is expected to depend on the cost of sexual reproduction [3]. The prediction is appealingly simple, but testing it encounters difficulties.

To cite four of those difficulties, it can be hard to measure plant "size," especially components such as diffuse underground root systems, without grave disturbance.

provided the original author and source are credited.

**Data availability statement:** Yes - all data are fully available without restriction; All data can be found at the following repository: https://doi.org/10.6073/pasta/e3e67039141f54c24c87326e5d15f49a.

**Funding:** The author(s) received no specific funding for this work.

**Competing interests:** That authors have declared that no competing interests exist.

Second, the theory does not concern plant size *per se*, but rather accumulated resource, for which size is merely a surrogate. The relationship between size and storage may be weak, for example in woody plants whose accumulated tissues may be massive but metabolically inert. Third, attack by natural enemies may obscure or obliterate the physiological relationships that link growth and reproduction. Fourth, investigations of truly long-lived perennials may require periods of observation that outlast typical investigators' research careers. Here, we report on a plant whose properties ameliorate the first three of these difficulties. As for the fourth, the study has already continued for more than thirty years and has a plausible potential to continue indefinitely. This paper's immediate goal is to address three issues of general interest to evolutionary ecologists: cost of fruiting, regulation of size, and senescence. The first two are easily answered, but the third will require continued data collection and new insights. Therefore, the longer-term goal is to advertise the data set. We hope to invite deeper analyses by others and to entice new investigators to take over the study. In addition to the annual records, measurements, and photographs of the corms, the data set archived at the Environmental Data Initiative (henceforth, "EDI archive") includes additional details on the project's history, methodology, R code [4], and summary tables and figures that are specifically relevant to this paper. Annual updates will be posted at https://portal.edirepository.org/nis/mapbrowse?scope=edi&identifier=574&revision=3

### The study plant

Like their familiar domesticated relatives the tulips, glacier lilies (*Erythronium grandiflorum* Pursh (Liliaceae)) annually shed their shoots and most of their roots to concentrate their substance into a discrete, starchy, overwintering corm (Fig 1). With care, these dormant resource-storage organs can be exhumed intact. Corm weight provides a quantifiable measure of plant size, analogous to but even simpler than Obeso's [5] counts of module production. Since 1993, we have followed numerous corms (now, 282) established in a transplant garden located on private property just east of Lake Irwin, Colorado (N38.8764830, W107.0993257, 3170 m elevation), about 13 km southwest of the Rocky Mountain Biological Laboratory (RMBL) in the West Elk Mountains. We record flower production in late May-early June, and fruit production in July. After the plants become dormant in mid-August, we exhume, weigh, and photograph the corms and then replant them.

Some aspects of local lily ecology have been reported. Predation of corms by pocket gophers (*Thomomys talpoides*) is frequent [6]. Thomson *et al.* [7] showed that it determines the distribution of flowering plants within habitats. Although seeds germinate best and plants grow well in pockets of deep soil, large flowering plants are concentrated in rockier soils and on rocky outcrops, where underground predation by gophers is hampered. Distributions are also affected by constitutionally poor seed dispersal [8]; unlike many *Erythronium* species, *E. grandiflorum* seeds lack elaiosomes that would foster dispersal by ants and wasps. It also lacks the ability for vegetative spread *via* stolons, so it does not form the large clonal patches that are characteristic of congeners such as *E. americanum* and *E. albidum* in eastern North America.

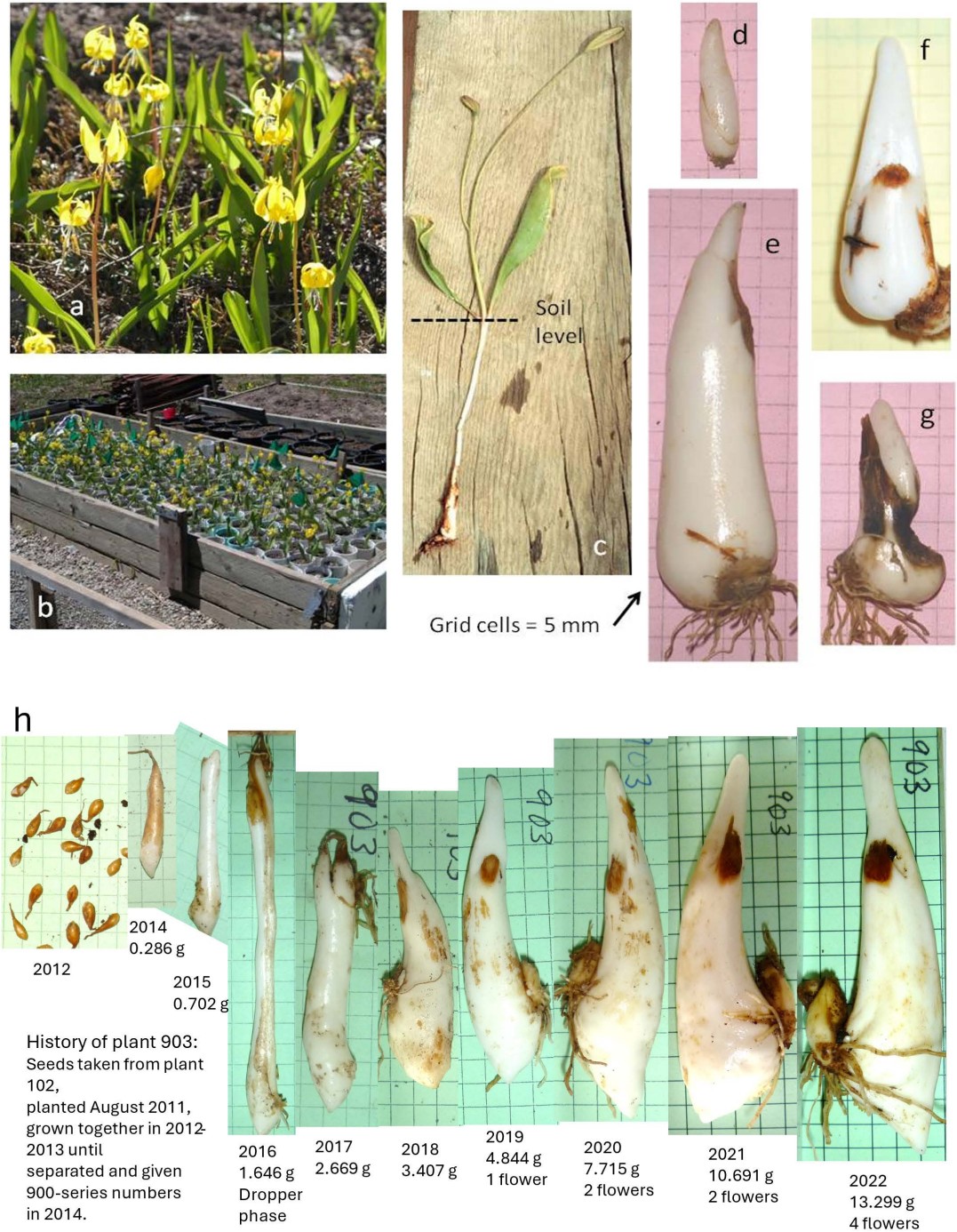

**Fig 1. Study plant and transplant garden. a.** *Erythronium grandiflorum* in flower. **b.** Transplant garden at Irwin, Colorado, USA, with plants in open-ended pots of PVC sewer pipe. **c.** Plant excavated in mid-season, showing immature fruits and deeply buried corm. **d.** Small healthy non-flowering corm. **e.** Large healthy flowering corm. **f.** Corm with incipient rot in cross-shaped surface crack. **g.** Large corm having survived rot but with much loss of mass. **h.** Progression of growth of one typical plant (903) from seedling to maturity. The 900-series plants were grown from sibling seeds planted communally in 2011, yielding teardrop-shaped first-year corms as shown in 2012. Twenty-one of these were planted singly in 2014 after they had reached about a quarter gram and begun to elongate. 903 spent 2015-2017 in dropper mode, reaching its extreme downward growth extension in 2016. From there onward, downward extension diminished as the corm stabilized in form and began producing flowers. From its peak in 2022, 903's weight dropped to 10.3 g in 2023 and 7.5 g in 2024 (not shown) after flowering and fruiting heavily.

In its first year of growth, a seedling of *E. grandiflorum* puts out a grass-like cotyledonary seed leaf. In subsequent years, it produces a single leaf with a broader blade is produced. Fig 1h shows the progress of a selected garden plant (903) grown from seed. (Pictures of all plants are in the EDI archive.) After a corm reaches a fresh weight of about 1–2 g, typically in 5–7 years, the plant switches to producing two leaves and a flowering scape. Underground, the plant remains near the surface as a teardrop-shaped corm for the first few years of growth. It then changes to a radically elongated "dropper" form, up to 10 cm long, in the process of translating its substance downward in the soil. A dormant dropper has two tufts of roots, an upper one from the previous summer and a lower one that will produce the following year's roots. After two or more seasons of "droppering", and reaching about 25 cm depth, a garden-grown plant reverts to the extended conical teardrop or "dogtooth" shape that gives the genus *Erythronium* one of its common names. In the field, however, large old corms are often distorted into complicated shapes because they have crammed themselves into cracks in rocks. Such situations provide refuge from gophers. Because droppers grow straight downward, the lateral spread of plants may occur primarily through transport of uneaten pieces by gophers traversing their underground tunnels [6]. Some plants in the field may therefore be highly vulnerable while near neighbors are securely protected. A plant wedged into a suitable rock crevice will experience similar protection to that of a plant in the study garden.

Most flowering plants in the field produce only a single flower, but in favorable locations, doubles are common, triples are occasional, and quadruples are rare. Blooming immediately after snowmelt, the earliest-flowering cohorts frequently fail to set fruit either through frost damage or inadequate pollination [9–12]. If a flowering plant is growing poorly, it may revert to producing a single leaf with no flower. Upon exhumation, a small number of plants have spontaneously split into two or three corms.

## Materials and methods, garden history

For reasons no longer relevant, we began with mature corms excavated on site. Subsequent cohorts were grown from seed, so their ages are known (details in EDI archive). The outdoor study garden, established in 1991, is a raised bed with lumber sides, now holding 282 open-ended 30 cm tall cylinders (henceforth, "pots") made of 10 cm diameter PVC sewer pipe (Fig 1b). The bottom of the garden is screened with galvanized wire mesh to exclude burrowing gophers. During the growing season, a removable cage of plastic "chicken wire" mesh excludes sciurids and cervids, but not flower-visiting insects. (Herbivory by invertebrates has always been inconsequential, probably due to the plant's brief and early growing season.)

To allow close contact between the pot contents and the soil below, we used galvanized wire mesh to make basket-like bottoms for the cylindrical pots. These allow the pot core to be extracted intact without damaging delicate corms (pictures at EDI archive). Annual exhumations of dormant corms would specifically allow tests of the (then) recently published predictions cited above [1,2]. As a secondary and broader question, following corms over many years might clarify whether and how these plants undergo physiological senescence. Therefore, we record flower numbers, fruit set (and seed set in some years). Every August, I record the fresh weights of exhumed corms, as well as (1) a subjective numerical score to represent the extent of "dropper" formation, (2) damage to corms, (3) occasional spontaneous splitting (or accidental breakage), and (4) evidence of fungal attack. We began photographing the exhumed corms in 1995. Spontaneous vegetative splitting occurs occasionally (e.g., 7 plants split in 2024, 2 in 2025). Some plants have never split during the study. Replanting split offsets in the same pot would result in confusing them, so each of the splits gets potted in its own pot. For example, when plant number 4 first split in 1992, the larger offset became 4A and the smaller one 4B. Some genotypes are more likely to split than others. These clones are now represented in the garden by numerous separate plants. (As of 2024, the clone with the most splits is in fact plant 4, whose eight surviving clonemates include 4ABAAAAAA and 4ABAAAAAB, which have registered 8 splits each. However, plant 43 has the most surviving clonemates (17), although none have experienced more than six splits. The difference between these two supersplitters is that splits of plant 43 tend to be more equal in size, while plant 4 tends to produce small offsets that are less likely to survive.

Pot positions in the garden are scrambled each year. Exhumation and repotting are done in ways that pool the soil from a "flight" of 20 pots, so that exposure to microbes and nutrients will be uniform. Corm death is rare. Very small corms occasionally disappear with no discernable cause. More often, remnants of dark brown rotted tissue and bits of mycelium point to a fatal fungal attack. Fungal attack can also be sublethal, leaving behind some chunks of healthy tissue (Fig 1g), as if part of the corm tissue had successfully walled off the infection. When replanted, these firm white nubbins frequently recover and continue life as new healthy splits. Most corms that suffer major trauma from fungal rot (*i.e.,* death or massive tissue loss) appeared healthy at the previous harvest.

## Results, part 1: Costs of fruiting and size regulation

### Growth patterns

After removing ambiguous records (*e.g.*, for corms damaged in handling), the data set used here includes 8551 annual transitions that collectively reveal how current life history events affect subsequent measures of performance. Young glacier lily seedlings do not flower until they are about 6 yr old and weigh over 1 g. For the garden population, logistic regression of flowering probability in spring of year N as a function of corm mass in autumn of year N-1 yields an inflection point at 1.08 g ($P < 2 \times 10{-16}$; distributions in Fig 2a). Having reached maturity, plants typically make two leaves and 1–4 flowers (Fig 2b). Flower number is positively related to prior corm weight ($P < 2 \times 10{-16}$; Fig 2c), and a mature plant can revert to a flowerless, single-leafed state if it shrinks enough. These responses allow plants to regulate body size. For a subset of plants started from seed, corm weight initially increased with age but leveled off after corms reached about 5 g (Fig 2d). The same pattern of regulation is also evident in the size histories of those corms that never split between 1996 and 2023 (Fig 3). The two highlighted lines in Fig 3 suggest that individual plants may regulate themselves around different targets. Those two plants consistently differ by about 3 g. The genetic basis of such differences could be investigated by future analyses of clonemates.

Weight histories of all plants that survived from 1996 through 2023 without splitting, showing a strong tendency toward size regulation, despite ups and downs. The darker lines show two trajectories selected to suggest that genotypes may regulate around different setpoints.

### Cost of fruiting

To account for inter-annual variation in growth, a case-control design evaluated the effect of fruit production during a summer on corm weight at the end of that summer (details and R code in the EDI archive). Producing a single fruit reduces expected growth by 8%. Relative growth declines with the number of fruits produced (evaluated by comparison to the average relative growth (in the same year) of non-fruiting plants most closely matched by weight, ($P = 0.025$; Fig 4)). In addition, fruit production reduces the number of flowers in the subsequent year. Comparing the number of flowers of fruiting plants to the average number of flowers of matched non-fruiting plants shows that the number of subsequent flowers declines with the number of fruits produced ($P = 0.0052$).

Mortality in the garden is highly dependent on corm size. Very small corms (< 0.5 g) are particularly vulnerable, probably because they are poorly buffered against stress. Once a corm is well established and flowering regularly, though, death is unlikely. Some corms lose weight gradually over several years, suggesting senescent decline (see below), but this is uncommon. More often, established corms either persist robustly or suffer from severe fungal attacks. Such attacks usually (but not always) leave obvious mycelial debris. Fig 5 shows the probability of trauma (defined as any of the three "bad outcomes": rot, extreme weight loss (> 75% of weight), or death) as a function of corm weight. Unsurprisingly, the probability of trauma drops as corm weight increases from small to medium. Surprisingly, it increases again as corms grow to extreme sizes beyond 5 g ($P = 4.32 \times 10{-12}$; Fig 5). Why larger corms become more susceptible to trauma is unclear. We speculate that rapid growth spurts may induce stress cracks in the surface, through which fungi accomplish infection. We have seen such cracks in some large corms (Fig 1f) but not in small or medium ones.

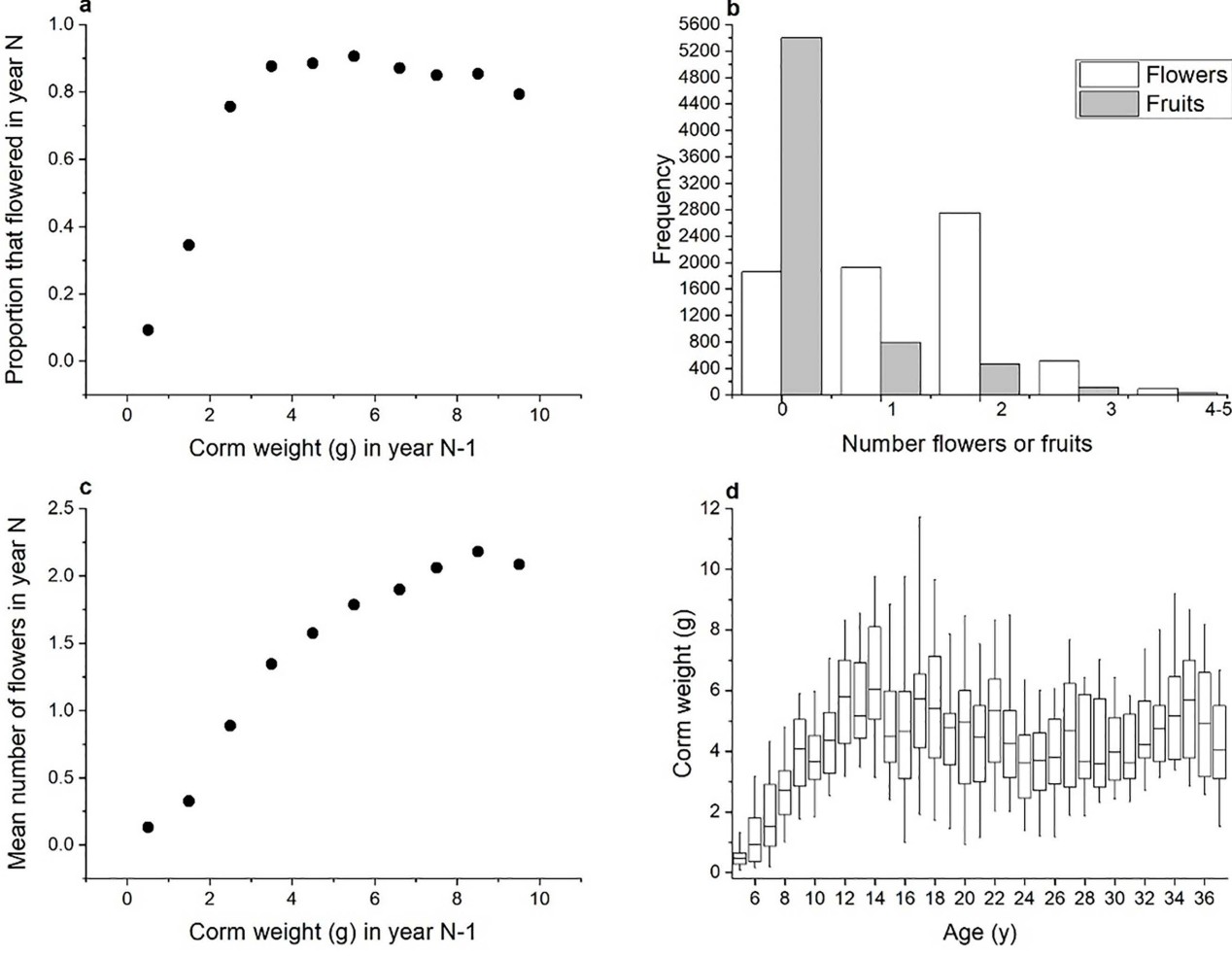

**Fig 2. Life-history characteristics of glacier lilies in the transplant garden. a.** Probability of flowering increases with corm size, with a transitional zone from non-flowering to flowering around between 1 and 2 **g.** The plotted points are proportions of many corms, divided into bins of 1 g width and plotted at the midpoint of the bin. Numbers in each bin, in order of increasing weight, are 851, 686, 968, 1291, 1028, 728, 459, 293, 189. **b.** Distributions of flower and fruit production in garden plants, pooled over all seasons. Distributions for plants in the field have similar shapes, but are shifted toward lower values; garden conditions are more favorable. **c.** Flower production increases with corm weight. Data are presented in bins of 1 g width, as in panel **a.** Numbers in each bin are 831, 628, 908, 1222, 1175, 983, 696, 429, 278, 174. **d.** For a representative set of 64 garden plants of known age, age and corm weight are correlated early in life as corms grow, but level off after 10-15 years. In established plants, weights are roughly regulated around 4-5 **g.** For plants that split, this data set retains only the largest offset.

## Discussion, part 1: Cost of fruiting and size regulation

Given the theoretical prediction that perennials should maintain an optimum weight, we expect to see regulatory mechanisms that would prevent overgrowth. Other *Erythronium* species, such as *E. albidum*, can prevent overgrowth by diverting spare resources into clonal expansion through stolons, but *E. grandiflorum* lacks that capability. Splitting does provide a safety valve, but in the field, the daughter ramets will remain in the same spot and therefore will compete with each other, so fitness gains would be smaller than they appear in the garden. The mechanism most likely to successfully avoid morbid overgrowth is the production of more fruits. This mechanism allows size regulation if ecological conditions allow fruit set, but such regulation is necessarily imprecise. If fruit set is consistently prevented by frost or inadequate

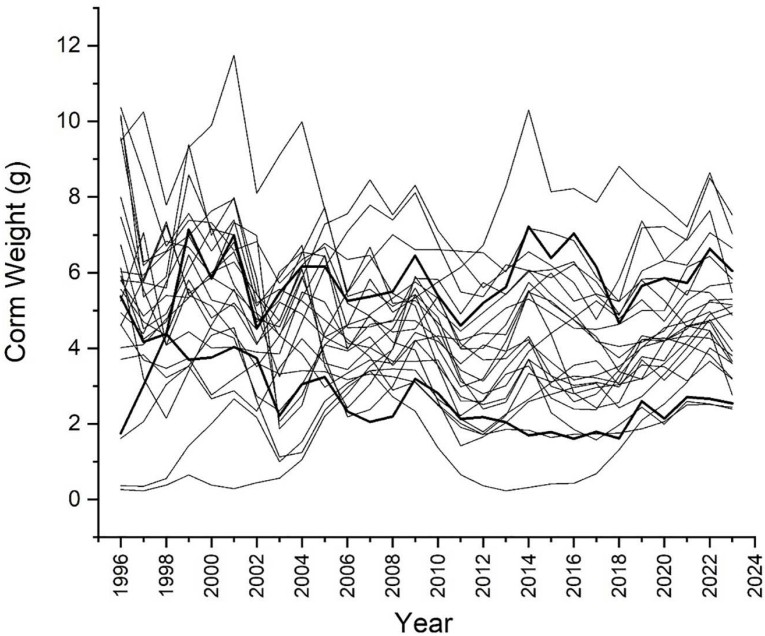

**Fig 3. Growth trajectories.**

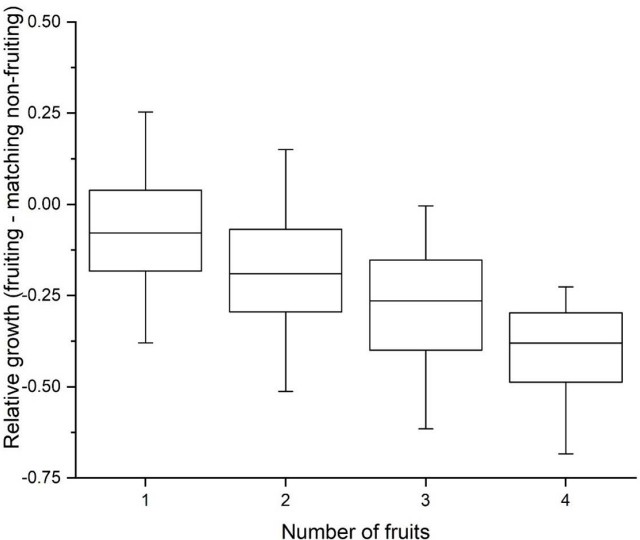

**Fig 4. Cost of fruiting.** Fruit production cuts into vegetative growth. Making a single fruit weight by about 8%, so a plant could shed as much as one-third of its weight by making four flowers, if all of them developed into fruits. More fruits cause more weight loss. This is a case control design in which the (log) relative growth of corms that fruited are compared to the relative growth of a matched set of nine non-fruiting corms of similar size in the same year. This matching removes the effects of calendar year and corm size from the comparison. Since corms with a weight < 1 g rarely fruit, only corms with a prior weight greater than 1 g were included in the comparison.

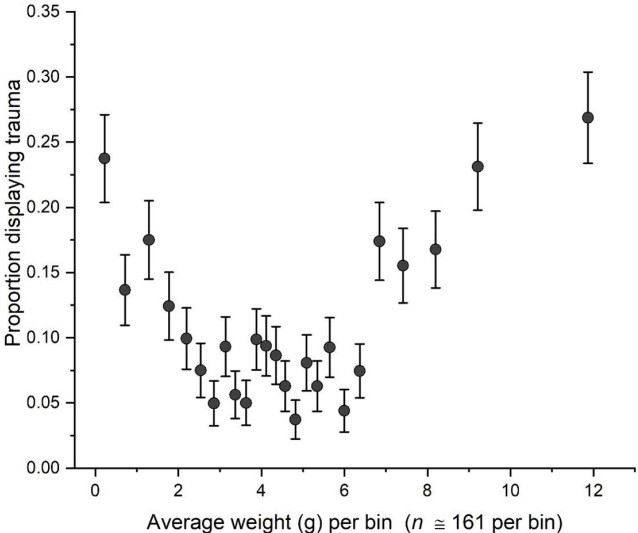

**Fig 5. Size dependence of damage to corms.** The probability of trauma (defined as death, a sudden single-year loss > 75% of corm weight, or evidence of rot from fungal attack) initially declines with increasing corm weight but increases again as corms become extremely large.

pollination, corms might grow to life-threatening sizes. This fate is currently unusual, but may become more common if climate change or other factors increase flower failure. Although a plant has limited control over how much weight it gains when fruiting fails, it does have the flexibility to spend resources heavily on fruit and seed production in seasons with good weather and pollination. Therefore, life-threatening weight gains might require several consecutive years of fruiting failure, but a single year of heavy fruiting might suffice to bring weight down to a healthy level. An experiment is underway to determine whether preventing fruiting every year causes plants to grow dangerously large. So far, non-fruiting plants have tended to respond by splitting rather than by growing excessively (data in EDI archive).

## Introduction, part 2: Senescence, splitting, and clonality

Although the garden data had answered the questions of size regulation and cost of fruiting within the first ten years or so, evaluating senescence will require longer data collection. Our hope in advertising the study is that younger researchers may be attracted to accept these challenges and to see the transplant garden as an opportunity for designing and conducting experiments. Here, we offer only a brief account of how the natural history of glacier lilies meshes with basic ideas about aging. The extent of clonal reproduction is pivotal.

Williams's foundational senescence theory [13] contrasted two sorts of organisms. In mortal unitary organisms, such as most animals, natural selection weakens with age. This allows the accumulation of mutations that are deleterious in old age, especially if those mutations are also advantageous in early life ("antagonistic pleiotropy"). This accumulation produces a genetically encoded senescence. In contrast, Williams [13] proposed that clonal reproducers, such as protozoa and some plants, would avoid such senescence. Clonality essentially confers immortality. A recent review [14] tends to support the prevalence of antagonistic-pleiotropy-based senescence in animals. Plants are more problematic. For example, Ally *et al.* present evidence of senescent decline in highly clonal aspen [15], where it would not be expected. Roach and Smith [16] reviewed 22 studies of plant senescence and 20 examples of life-history tradeoffs in plants, concluding (p. 17), "We found all three age-specific patterns: age decline, age improvement, and change with age for plants in the wild. Most species showed an age-dependent decline in at least one trait but, even within a species, the age-specific patterns of traits varied." Our *Erythronium grandiflorum* plants include some genotypes that have not reproduced clonally

in 30 years and others that have split repeatedly. But it seems premature to characterize lily populations as mixtures of unitary mortals and clonal immortals.

Although our data cannot support a definitive yes/no evaluation of senescence, we can make a start by examining and classifying the growth trajectories of plants in the years before death. Importantly, plant fates in the garden will differ from those in the field, where gopher predation is evidently important. Nevertheless, if plants display "classic" senescence, *i.e.*, if antagonistic pleiotropy has caused the accumulation of numerous deleterious genes that act late in life, the resulting deterioration should occur whether predation is prevented or not.

## Results, part 2: Growth trajectories

The growth trajectories of individual corms can address two specific questions. First, do corms gradually decline to death over several years? As mentioned above, such trajectories are rare, but they do conform to classic senescence due to accumulating declines in vigor. Second, given the unexpected pattern (Fig 5) that oversized corms show elevated death rates, are those deaths associated with *gradual* growth to great size or with rapid growth spurts? The former pattern would suggest that more resources are detrimental, which seems perverse. The latter would be more consistent with the hypothesis that sudden growth spurts might induce surface cracks that invite fungal attack.

The answers depend on examining many growth trajectories. Although too voluminous to present here, the EDI archive (under the heading "Senescence") presents two sets of graphs of weight versus time for 148 plants that—as of the 2024 harvest—met the simplifying criterion of not having split in the last seven years of their lives. Fig 6 shows two of these trajectories, chosen to represent very different end-of-life histories: the gradual, progressive deterioration of "classic senescence" *versus* sudden collapse.

In the first analysis of the 148 trajectories (termed "Slopes" in the EDI archive), a linear regression is fit through the first six of the seven years. A negative slope indicates that the corm has been losing weight, and the trajectories are sorted by slope, six trajectories per graph. In those graphs, the fate of the corm is indicated by line color: black = survived without incident, green = rotted but survived, blue-purple = death with rot, and red = died without evident rot. The trajectories are highly

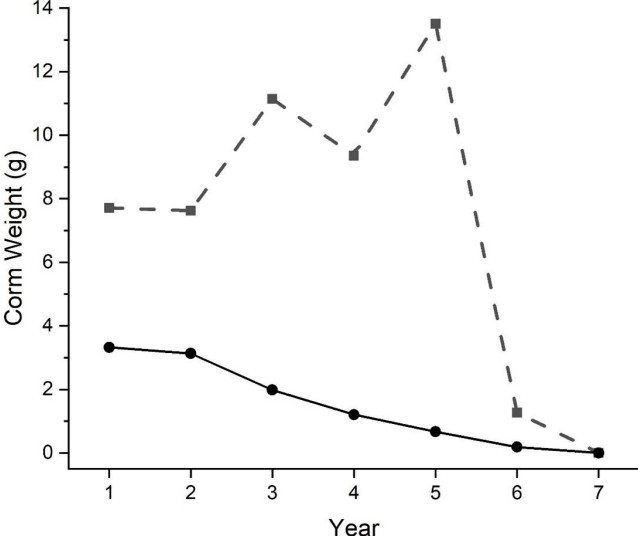

**Fig 6. Two pre-death growth trajectories, selected to exemplify alternative routes to death.** The lower curve conforms to "classical senescence," in which a corm gradually dwindles as if from deteriorating vigor. in contrast, the upper curve shows a seemingly healthy corm that was growing rapdly before a sudden collapse. The full set of trajectories from which these two were selected can be found in tje online archive.

individualistic and should be examined as a collection. However, there are a few cases of progressive decline to death without rot, *i.e.,* red lines with a consistently negative slope leading up to death. Like the lower line in Fig 6, cases are consistent with a smooth, unremarkable senescence in which an organism gradually dwindles away due to diminishing vigor.

In the second analysis of the 148 trajectories, termed "Spurts," a linear regression is fit as above, but then the largest single-year weight deviation above the line is identified. The intention is to capture episodes of sudden excess growth, to determine whether such growth spurts do increase vulnerability to fungal attack, as hypothesized above. In the Spurts section of the archive, trajectories are sorted by the size of the maximum deviation.

## Discussion, Part 2

To distill the complicated relationship between weight trajectories and corm fate, Fig 7 plots Slope versus Spurt for the 148 trajectories, with separate panels for different corm fates. The most common fate—continued survival with no

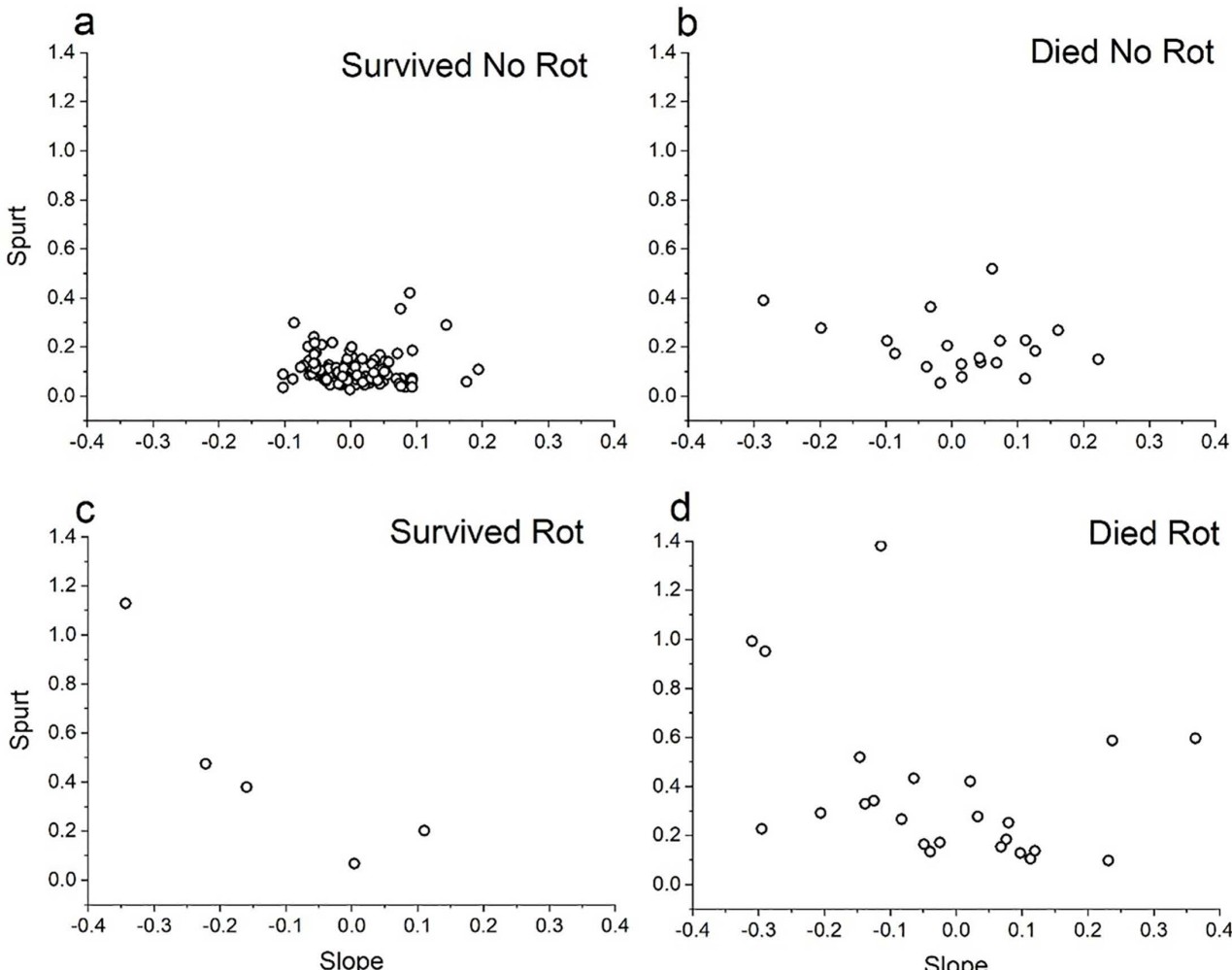

**Fig 7. Growth trajectories and corm fates.** Relationship of non-splitting corm fates to the trajectory of weight change over the previous six years, as summarized by linear regressions. "Slope" indicates the slope of the fitted regression line, with steeper declines having higher values. "Spurt" indicates the highest single-year deviation from the fitted line. Corms that survive intact and free from fungal infection have moderate slopes and small spurts. These exemplify "classic senescence." Fungal attack is more likely in corms that have undergone growth spurts, consistent with the pattern in Fig 5. These deaths differ from classic senescence in that plants were growing vigorously shortly before dying.

evidence of rot—is associated with shallower slopes and smaller spurts; these trajectories form a tight central cluster (panel a). In contrast, corms that experienced fungal attack (panels c and d) were more likely to have undergone growth spurts (points displaced upward) and to show a greater range of slopes (wider spread of points). Corms that died without leaving traces of rotted tissue (panel b) also showed a wider range of slopes. This group includes some small corms for which fungal attack might not leave detectable traces. Overall, this analysis (in conjunction with the plotted trajectories in the archive) suggests that (1) growth spurts do predispose corms to fungal attack and (2) death from spurting and rotting is more common than death from slow decline. Continued study for more seasons will be necessary to clarify these relationships. Death triggered by growth spurts will probably be elicited by three factors: low fruit set due to frosts or lack of pollination; favorable weather for photosynthesis and vegetative growth; and favorable conditions for fungal growth. Unless all three of these conditions co-occur, most corms will be expected to persist without incident.

Table 1 summarizes the prominent patterns within the garden. The salient observation is that trajectories consistent with "classic" senescence are rare. The biggest outstanding question is whether they will become more common as the corms age further.

## Conclusion

In a transplant garden that prevents predation, corms of *Erythronium grandiflorum* regulate their weight through several physiological responses. Larger corms make more flowers and therefore more fruits, on average. Producing fruit reduces corm weight. (Plants might also regulate their costs of reproduction by aborting fertilized flowers, or by adjusting the numbers of seeds per fruit, although these mechanisms have not been studied here.) In addition, large corms may reduce overgrowth by vegetative splitting.

Observations from field and garden suggest that some individuals within *Erythronium grandiflorum* populations are living as "unitary" organisms like those typical of highly mortal animals. Others may attain near immortality by insinuating themselves into predation-free microsites. Still others may achieve near immortality through clonal proliferation. Deriving expectations for the evolution of senescence in such a population will require further modeling, ideally underpinned by extended data on individual life histories. It remains possible that many of the genotypes in the garden will eventually succumb to "classic" senescent declines, although only a few have done so already. Extending the study could answer this question.

**Table 1. Summary of the status of *Erythronium grandiflorum* plants at the end of seven years' growth without splitting.**

| Corm fate | Category of death | Occurrence in transplant garden | Occurrence in the field | Desirable further study |
|---|---|---|---|---|
| Continued survival without splitting | -- | Common | Probably common | Extend observations |
| Continued survival with splitting | -- | Very common | Probably common | |
| Death by predation | Accidental, not senescence | Prevented by design | Common | Observe isolated plants in field? |
| Death of vigorous plant by failure to prevent overgrowth, leading to trauma | Accidental, not senescence | Frequent in certain years and conditions | Unknown, probably less frequent than in garden | Extend observations to specify required combinations of conditions |
| Death by loss of vigor, shown by gradual decline in growth | Classic senescence by breakdown of physiological competence | Rare so far (5 out of 148 cases) | Unknown, probably more frequent than in garden | Extend observations, compare clone mates |

## Acknowledgments

Barbara Thomson provided essential help at every stage. David Inouye, Jennifer Reithel, and Samantha Williams helped at critical points.

## Author contributions

**Conceptualization:** James D Thomson.

**Formal analysis:** David F. Andrews.

**Investigation:** James D Thomson.

**Methodology:** James D Thomson.

**Writing – original draft:** James D Thomson.

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
