## [Decision Letter · Decision Letter 0]

8 Oct 2025

PONE-D-25-47577Cost of fruiting, size regulation, and senescence in a long-lived geophytePLOS ONE

Dear Dr.  Thomson,

Thank you for submitting your manuscript to PLOS ONE. After careful consideration, we feel that it has merit but does not fully meet PLOS ONE’s publication criteria as it currently stands. Therefore, we invite you to submit a revised version of the manuscript that addresses the points raised during the review process.

We look forward to receiving your revised manuscript.

Kind regards,

Juan Carlos Suárez Salazar

Academic Editor

PLOS ONE

Journal Requirements:

When submitting your revision, we need you to address these additional requirements

2. We noted in your submission details that a portion of your manuscript may have been presented or published elsewhere ”Data have been placed in a public archive. Thomson, J.D. 2025. Life histories of the perennial geophyte Erythronium grandiflorum (Liliaceae) in Colorado subalpine transplant garden from annual measurements, 1991 onward ver 3. Environmental Data Initiative. https://doi.org/10.6073/pasta/e3e67039141f54c24c87326e5d15f49a” Please clarify whether this [conference proceeding or publication] was peer-reviewed and formally published. If this work was previously peer-reviewed and published, in the cover letter please provide the reason that this work does not constitute dual publication and should be included in the current manuscript.

3. Thank you for uploading your study's underlying data set. Unfortunately, the repository you have noted in your Data Availability statement does not qualify as an acceptable data repository according to PLOS's standards.

Additional Editor Comments:

Following an exhaustive review of the manuscript on the life cycle of Erythronium grandiflorum by three reviewers, crucial aspects requiring attention have been identified to significantly improve the scientific quality and impact of the work. The current manuscript, while containing valuable data based on years of research, needs substantial restructuring to meet scientific publication standards.

The primary concern focuses on the document's structure, which currently resembles an essay more than a formal scientific article. It is essential to incorporate clearly defined sections for Methodology, Statistical Analysis, and Discussion. Results and discussion should be separated to improve clarity and logical flow. The writing style should remain consistent, using past tense for reporting results and maintaining a formal tone throughout the document.

From a technical perspective, the manuscript would benefit significantly from greater integration with recent studies on physiological and biochemical mechanisms. It is particularly important to relate the observed cost of fruiting (8% reduction in corm weight per fruit) to changes in photosynthetic pigments, antioxidant enzyme activities, and hormonal balance, especially concerning gibberellic acid and cytokinin. Including the suggested reference from Khan and Nabi (2024) would strengthen this connection between morphological results and underlying biochemical mechanisms.

A crucial aspect requiring clarification is the distinction between identified senescence pathways: gradual decline versus sudden collapse due to fungal attack. It is recommended to include biochemical or anatomical markers to support claims about "classical senescence." Additionally, the interpretation of "fungal attacks" should be approached with caution, considering that artificial study conditions might have altered soil microorganism balance.

Regarding visual presentation, figures need improvement, especially Figure 5, which requires confidence intervals and clearer labels, and Figure 6, where plants corresponding to gradual versus sudden death should be specified. Terminology should be consistent throughout the document, avoiding colloquial phrases and maintaining precise scientific language.

The manuscript should also address climate change implications, considering how changing frost patterns and pollination windows could affect the cost-benefit balance of reproduction. The importance of long-term monitoring in validating evolutionary life history theories should be highlighted, citing fundamental works such as Williams (1957) and Obeso (2002).

Finally, a critical revision of redundant or unnecessary sections is recommended, retaining only those examples and discussions that directly contribute to the main scientific argument. Methodology should be described concisely, referencing previous work where appropriate, and new data should be clearly differentiated from previously published results.

With these modifications, the manuscript has the potential to become a significant contribution to the field, making the most of the valuable dataset compiled during years of dedicated research.

Reviewer's Responses to Questions

**Comments to the Author**

1. Is the manuscript technically sound, and do the data support the conclusions?

Reviewer #1: No

Reviewer #2: Yes

Reviewer #3: Yes

2. Has the statistical analysis been performed appropriately and rigorously? 

Reviewer #1: No

Reviewer #2: Yes

Reviewer #3: I Don't Know

3. Have the authors made all data underlying the findings in their manuscript fully available?

Reviewer #1: No

Reviewer #2: Yes

Reviewer #3: Yes

4. Is the manuscript presented in an intelligible fashion and written in standard English?

Reviewer #1: Yes

Reviewer #2: Yes

Reviewer #3: No

5. Review Comments to the Author

Reviewer #1: The manuscript submitted for review focuses on the fascinating subject of the life cycle of Erythronium grandiflorum. The author has devoted many years to this topic and has already published several papers on it. Research into the biology of species is very difficult and often requires special, 'original' methodology. Unfortunately, interpreting the data is not easy. While the manuscript undoubtedly contains valuable data, it requires thorough rewriting and adaptation to the editorial requirements of PLOS ONE.

The present form of the text is more in the style of an essay than a professional scientific paper. A section entitled 'Materials and Methods' should be added that concisely describes the research methodology without going into too much detail and refers to previously published works in which I believe the methodology was the same. Many of the paragraphs in the paper are digressions and should be included in the Discussion section. I am also interested in how the data presented in this manuscript differs from the results of previously published studies conducted over a shorter period of time.

I'm wondering whether the longer research period allowed for new data to be established, or if it simply confirmed what was already known. I would also like to suggest to the author that he interprets the results of his research into ‘fungal attacks’ with a little more caution, given that he conducted his research under artificial conditions, which could have disturbed the balance of soil microorganisms. In their natural state, soil microorganisms exist within a balanced ecosystem that comprises both fungi and bacteria. Some of these microorganisms, such as bacteria, are known to stimulate plant growth and development; however, under disturbed conditions, they can cause plants to become weak and die. Some bacteria can increase a plant's susceptibility to fungal infections. Since the author exhumed the corms, he may have affected their ‘quality of life’ in this way. I would therefore exercise caution when interpreting the obtained results.

Minor:

Lines 51–52: This sentence is unnecessary.

Lines 56–57: Is this sentence necessary?

Lines 94–102: This section should be shortened and the digressions omitted.

Lines 117–128: Is this section necessary?

Lines 144–149: As above, are these examples necessary?

Reviewer #2: ***The discussion could benefit from broader integration with recent studies that examine physiological and biochemical mechanisms underlying fruiting costs and senescence.

Please relate the observed cost of fruiting (8% reduction in corm weight per fruit) to changes in photosynthetic pigments, antioxidant enzyme activities, and hormonal balance (especially gibberellic acid and cytokinin), which are known to influence resource allocation and longevity.

*** Strongly recommended citation:

Khan, M. N., & Nabi, G. (2024). The potential role of gibberellic acid in regulating photosynthetic pigments, fruit quality and antioxidant enzymes in sweet lime Citrus limetta Risso. Pakistan Journal of Botany, 57(2).

This paper would help the authors connect morphological outcomes (size regulation, senescence) to underlying biochemical mechanisms (GA-mediated regulation, oxidative stress mitigation), offering a more holistic perspective on plant life-history trade-offs.

***The paper suggests two senescence pathways (gradual decline vs. sudden collapse due to fungal attack). Please differentiate physiological senescence from pathological death more explicitly.

If possible, consider including or referencing biochemical or anatomical markers (e.g., tissue browning, ROS accumulation, chlorophyll degradation) to strengthen claims of "classical senescence."

***Ensure Figure 5 (probability of trauma vs. corm weight) is clearly labeled and includes confidence intervals or error bars.

In Figure 6, specify which plants correspond to gradual vs. sudden death to guide interpretation.

Please confirm that all figures referenced in the text (Figs. 1–7) are included and legible.

***Replace phrases like “plants might grow themselves to death” with more formal wording such as “plants may experience mortality due to overaccumulation of biomass and subsequent vulnerability to infection.”

Ensure consistent use of terms like “cost of fruiting” vs. “reproductive cost.”

***Line 17–19: consider rephrasing to “producing one fruit results in approximately 8% reduction in potential corm weight gain.”

Line 199–201: clarify speculative statements with references or indicate explicitly as a hypothesis.

***Briefly discuss potential climate change implications, as shifting frost patterns and pollination windows may alter the cost-benefit balance of reproduction.

Mention the importance of long-term monitoring in validating evolutionary life-history theories (Williams 1957, Obeso 2002).

Reviewer #3: The manuscript titled "Cost of fruiting, size regulation, and senescence in a long-lived geophyte" addresses a very interesting topic—the study of senescence in plants. The main issue with the current version is that its structure does not conform to a standard scientific format. It lacks dedicated sections for Methods, Statistical Analysis, and Discussion. Furthermore, the statistical analyses need to be described more clearly. Currently, the results and discussion are intertwined, which reduces the clarity of the presentation.

Additionally, the writing style should be made consistent. The reporting of results should be in the past tense, whereas the current narrative often resembles informal notes rather than a formal scientific paper.

Despite these issues, the paper has significant potential but requires substantial reorganization and stylistic refinement to meet the standards of a scientific publication.

6. PLOS authors have the option to publish the peer review history of their article (what does this mean?). If published, this will include your full peer review and any attached files.

Reviewer #1: No

Reviewer #2: No

Reviewer #3: No

---

## [Author Response · Author response to Decision Letter 1]

18 Mar 2026

As requested by you, this issue is covered in the separate file "Response to Reviewers."

---

## [Editor Report · Decision Letter 1]

26 Mar 2026

Cost of fruiting, size regulation, and senescence in a long-lived geophyte

PONE-D-25-47577R1

Dear Dr. James D Thomson,

We’re pleased to inform you that your manuscript has been judged scientifically suitable for publication and will be formally accepted for publication once it meets all outstanding technical requirements.

Kind regards,

Juan Carlos Suárez Salazar

Academic Editor

PLOS One

Additional Editor Comments (optional):

Most editor and reviewer comments were addressed or explicitly clarified in the revision. The manuscript now includes clearly labeled Materials and Methods, Results (Part 1 / Part 2), and Discussion sections. All statistical analyses were designed and coded by David F. Andrews, now a co-author; the R code and additional output are archived in the EDI repository and are referenced in the data files. The author provided the dataset DOI/URL at the Environmental Data Initiative and confirms that EDI is an acceptable repository; a minimal, publicly accessible data subset was also uploaded. The author explains that no physiological data exist and that collecting such measurements would compromise the demographic time series, so those experiments are outside the study’s scope. The manuscript expands the discussion that distinguishes gradual decline from sudden fungal collapse, directing readers to the growth-trajectory graphs in the EDI archive and acknowledging the study’s limitations. Figure captions and references were improved. The author defends the omission of confidence intervals for Figure 5 as inappropriate for that analysis and reports inability to upload PACE-processed figure files due to account problems. The text has been revised for clarity and consistency (terminology and verb tense), and suggested citations were evaluated and added where relevant. Climate-change implications are mentioned, but the author reserves detailed analysis for future collaborative work. In summary, the author has documented responses to most formal requests concerning structure, data, and statistics (including adding the statistician as co-author and depositing data). Requests that would require new data could not be fulfilled and are reasonably justified. A few operational tasks remain incomplete due to account issues; the author requests editorial flexibility on those points.
---

## [Editor Report · Acceptance letter]

PONE-D-25-47577R1

PLOS One

Dear Dr. Thomson,

I'm pleased to inform you that your manuscript has been deemed suitable for publication in PLOS One. Congratulations! Your manuscript is now being handed over to our production team.

Kind regards,

on behalf of

Dr. Juan Carlos Suárez Salazar

Academic Editor

PLOS One